# A Fast-Tracking-Particle-Inspired Flow-Aided Control Approach for Air Vehicles in Turbulent Flow

**DOI:** 10.3390/biomimetics7040192

**Published:** 2022-11-06

**Authors:** Hengye Yang, Gregory P. Bewley, Silvia Ferrari

**Affiliations:** Sibley School of Mechanical and Aerospace Engineering, Cornell University, Ithaca, NY 14853, USA

**Keywords:** flight control, turbulent flow, fast-tracking effect

## Abstract

Natural phenomena such as insect migration and the thermal soaring of birds in turbulent environments demonstrate animals’ abilities to exploit complex flow structures without knowledge of global velocity profiles. Similar energy-harvesting features can be observed in other natural phenomena such as particle transport in turbulent fluids. This paper presents a new feedback control approach inspired by experimental studies on particle transport that have recently illuminated particles’ ability to traverse homogeneous turbulence through the so-called fast-tracking effect. While in nature fast tracking is observed only in particles with inertial characteristics that match the flow parameters, the new fast-tracking feedback control approach presented in this paper employs available propulsion and actuation to allow the vehicle to respond to the surrounding flow in the same manner as ideal fast-tracking particles would. The resulting fast-tracking closed-loop controlled vehicle is then able to leverage homogeneous turbulent flow structures, such as sweeping eddies, to reduce travel time and energy consumption. The fast-tracking approach is shown to significantly outperform existing optimal control solutions, such as linear quadratic regulator and bang-bang control, and to be robust to changes in the vehicle characteristics and/or turbulent flow parameters.

## 1. Introduction

Animals such as soaring birds, migrating insects, and swimming fish can traverse turbulent flows efficiently by taking advantage of approximately stationary flow structures [1,2,3,4,5,6]. Birds such as eagles and storks with large wing spans and surface areas are able to detect and exploit rising thermals or shear flows to generate lift and therefore save energy for long-distance flight [7,8,9]. Migrating insects can adaptively change their headings to harvest energy from atmospheric structures and motions based on their real-time measurements from wind-sensitive hairs and antennas [10,11,12,13]. Fish are found to be capable of detecting their surrounding flow features using the lateral line flow sensory system, and learn to adjust their swimming speed and body undulation while traversing turbulent water currents [14,15,16,17]. Many of these energy-harvesting features discovered in animal flyers and swimmers have also been observed in the characteristic motions of particles and bubbles carried by turbulent flows [18,19,20,21], which have inspired the new flow-aided air-vehicle feedback control design presented in this paper.

There is significant precedent for tackling air-vehicle navigation and control problems in strong but constant winds [22,23,24] and thermals [25]. Despite the prevalence of turbulence, its impact on locomotion, and the potential inherent in its energetic yet organized internal structure [20,26], most existing approaches either treat wind effects as disturbances to be rejected or require global knowledge of the entire wind velocity field [27,28]. This global knowledge may be acquired through learning [29,30,31,32] or with environmental prediction, modeling, and forecasting [33]. For instance, in [25] global knowledge is acquired by simulating turbulent thermals similar to those arising in the atmospheric boundary layer, and by using model-free reinforcement-learning algorithms to train gliders to soar. Besides requiring prior training, this approach generates more conservative policies than those observed in piloted gliders, and requires gathering information about the fluctuating flow while simultaneously ascending in it. Another approach is to exploit globally known flow structures produced by environmental prediction and forecasting algorithms to generate optimal vehicle trajectories using methods such as mathematical programming, differential evolution, or Lagrangian coherent structures (LCSs) [33,34,35]. While this approach is useful for underwater vehicles because ocean currents may be predicted to some extent using oceanographic modeling and prediction tools [36,37,38,39,40,41], it is less suited to air vehicles that must navigate rapidly changing winds without knowledge of global turbulent structures [42,43,44].

The process of particle transport in turbulence demonstrates that under certain conditions inertial particulates and droplets move quickly through turbulent flows such as turbulent air, water, or flames, without global knowledge of the velocity field [21,45,46,47]. The fast-tracking effect is the phenomenon by which inertial particles in turbulent flows exhibit an average settling velocity that is larger in turbulence than in still air [18,48]. Fast tracking of particles and droplets has been observed and verified in both physical experiments [48,49], and direct numerical simulations (DNS) of the gravitational settling of inertial particles in complex flow fields, including cellular flow fields [18], Gaussian random flow fields [50], and homogeneous isotropic turbulence [49,51]. Toward the exploitation of this phenomenon, Ref. [21] analyzes theoretically the energetics of idealized fast-tracking flight vehicles that make only local, instantaneous measurements, revealing an extended parameter regime in which turbulence can decrease flight time or energy consumption in principle.

This paper presents a new feedback control approach inspired by turbulent particle transport theory [21] that is able to reproduce fast tracking in air vehicles traversing turbulent flow fields. By viewing the particle dynamics as the ideal response to the surrounding flow, implicit model following (IMF) can be used to design a fast-tracking control (FTC) system that, by virtue of the onboard propulsion and actuation, induces the vehicle to behave like a particle in the closed loop. As a result, the vehicle flies within advantageous tailwinds more often than with existing control methods. The vehicle also avoids adverse headwinds automatically, thereby reducing the energy and time required to traverse a turbulent flow, and it does so without access to global flow information. The energy-harvesting potential of the new FTC control approach is demonstrated through two benchmark control problems known as the minimum-energy and minimum-time problems. The FTC-controlled vehicle performance is compared to two optimal control solutions obtained using linear-quadratic regulator (LQR) and bang-bang control (BBC) theories. The LQR solution to the minimum-energy problem is derived by using information about the flow field to make the vehicle reach and maintain a desired steady-state velocity using minimum control effort. The BBC solution to the minimum-time problem is derived by making the vehicle reach the final desired position in minimum time in still fluid.

Although the FTC approach only requires instantaneous knowledge of vehicle state and local flow, which are easily obtained onboard, it significantly outperforms both LQR and BBC designs. This general approach to flow-aided feedback control can also be applied to other vehicles including fixed- or flapping-wing aircraft [52,53], rotorcraft, and neutrally buoyant vehicles such as submarines or balloons, and to non-stationary flow structures such as thermal updrafts or mean shear [1,54]. The primary advantage of the fast-tracking approach over existing methods is that unsteady turbulent structures can be leveraged without relying on any prediction of the velocity field. In fact, by drawing inspiration from nature, the FTC design only requires approximate knowledge of a few key flow parameters, such as the mean velocity, the typical vortex length scale, and the typical vortex timescale, which can be easily estimated onboard, as shown in [55]. Extensive numerical simulations show that an air vehicle using FTC follows the ideal response of the fast-tracking particle with zero tracking error, regardless of its true inertial characteristics. As a result, the FTC system increases the average horizontal velocity of the vehicle, maintains the desired steady-state velocity with less control effort than the LQR solution, and reaches a desired horizontal position before the BBC solution.

This paper is organized as follows. Section 2 reviews relevant background from transport theory on the fast-tracking effect and cellular flows used here for illustrative purposes. The fast-tracking feedback-control design problem is formulated in Section 3, along with its basic assumptions. The FTC control design solution derived using implicit model following is presented in Section 4. In Section 5, the FTC energy-harvesting ability is demonstrated by comparing its performance to that of the optimal LQR solution on a benchmark minimum-energy control problem. In Section 6, the FTC time-saving ability is demonstrated by comparing its performance to that of the optimal BBC solution on a benchmark minimum-time control problem. Finally, the FTC performance robustness with respect to the vehicle inertial characteristics and turbulent flow parameters is demonstrated through dozens of representative case studies in Section 5 and Section 6.

## 2. Background on Transport Theory and the Fast-Tracking Effect

Natural phenomena such as soaring birds exploiting thermal convection [2] and particles or bubbles in steady vortices [19] demonstrate that turbulent air flows can be traversed rapidly and efficiently by leveraging local knowledge of approximately stationary flow structures. In particular, the mechanism known as the “fast-tracking effect” has been shown to govern the fast and intelligent motion of inertial particles in homogeneous turbulence [2,49,50,56]. This paper presents an approach for using the fast-tracking particle dynamic model, known from transport theory, in order to develop high-performance feedback control laws that can be implemented on autonomous vehicles in turbulent flow using local wind measurements and classic state estimation algorithms. The approach is demonstrated for a cellular-flow homogeneous-turbulence model, described in Section 2.2, which has been shown effective at capturing vortical structures in natural flows relevant to autonomous vehicles, such as atmospheric circulation [57] and ocean currents [58,59].

### 2.1. Fast-Tracking Effect

Experimental studies have shown that fast-tracking particles in a turbulent flow field are preferably thrown out of vortices, toward their downward-sweeping sides (Figure 1a), through a mechanism that increases the average speed of particles toward the bottom of the flow [49]. More precisely, the fast-tracking effect causes the mean settling velocity of an inertial particle traversing a turbulent flow to be increased with respect to the still-fluid settling velocity [49]. The particle’s mean settling velocity, denoted by *v*, is the average falling speed of particles subject to drag and gravitational forces when reaching an average force balance in a turbulent flow, or,
(1)$$v=\frac{1}{t_{f}-t_{s}}\int_{t_{s}}^{t_{f}}\;v_{p}\left(t\right)\;dt,$$
where $v_{p}$ is the particle’s instantaneous velocity, $t_{s}$ is the settling time required for the velocity to reach and remain within a given error band, and $t_{f}$ is the terminal time [60]. The still-fluid settling velocity, denoted by $v_{g}$, is the terminal falling speed of a particle through a still fluid. Therefore, when fast tracking prevails, it can be observed that $v>v_{g}$.

As shown in [49], fast-tracking particles are characterized by physical characteristics, such as response time and settling velocity, that “resonate” with those of the turbulent flow, as reviewed in the remainder of this subsection. Let *m* and *D* denote the mass and diameter of the particle, respectively. Then, when the particle is surrounded by a fluid with dynamic viscosity μ, its motion is characterized by the inertial response time,
(2)$$\tau ≜\frac{m}{3\pi D\mu }$$
as shown in [61]. In particular, the inertial response time represents the time required to reach equilibrium in response to perturbations in surrounding flows. The nature of the interaction between the particle and the flow depends on τ as well as on the characteristics of the turbulent flow, namely, the root-mean-square fluid velocity $u^{\prime }$, the vortex length scale *l*, and the vortex time scale $\tau_{w}\equiv l/u^{\prime }$ [49]. The root-mean-square velocity, $u^{\prime }$, is defined as the standard deviation of the instantaneous flow velocity, *u*, such that [62]
(3)$$u^{\prime }=\sqrt{\frac{1}{T}\int_{0}^{T}\;{\left[u\left(t\right)-\bar{u}\right]}^{2}\;dt},$$
where $\bar{u}$ is the mean flow velocity over the time interval *T*.

When the particle’s inertial response time (τ) and the still-fluid settling velocity ($v_{g}$) approach the turbulent flow’s vortex time scale ($\tau_{w}$) and the root-mean-square velocity ($u^{\prime }$), respectively, the particle’s settling velocity is significantly increased compared to its still-fluid settling velocity. Recently, this fast-tracking effect has been demonstrated experimentally by co-author Bewley using water droplets settling in air turbulence, as shown by the data plotted in Figure 1b and taken from [49]. In particular, this study showed that the (normalized) increase in settling velocity,
(4)$$△\hat{v}≜\left(v-v_{g}\right)/u^{\prime }$$
is positive whenever the particle undergoes the fast-tracking effect.

Consider the dimensionless particle-settling parameter, $\eta_{v}≜v_{g}/u^{\prime }$, which governs the onset of fast-tracking observed when $\eta_{v}$ is of order one. When $\eta_{v}<<1$ or $\eta_{v}>>1$, the particle’s mean turbulent settling velocity (*v*) is not enhanced compared to $v_{g}$. In fact, a sharp decline of the settling velocity is observed for large particle settling parameter, $\eta_{v}$, due to the development of nonlinearity in drag forces on quickly settling particles. Importantly, the normalized increase in settling velocity, $△\hat{v}$, is maximum when $\eta_{v}$ is of order one and, therefore, this parameter value can be used as a guiding principle in the development of a feedback controller that leverages turbulent flow to accelerate the vehicle similarly to the fast-tracking particle in Figure 1a.

This paper develops a new control approach by viewing the autonomous air vehicle in a turbulent flow as an inertial point-mass particle driven by a constant horizontal thrust that can be adjusted so as to match the desired fast-tracking characteristics of the given turbulent flow. There is considerable precedence for treating vehicles as point masses for navigation and control purposes, whenever their size is small relative to the vortex length scale [44,63,64,65,66,67,68,69]. Hence, our hypothesis is that by producing a controlled thrust that modifies the vehicle’s inertial response time to match the vortex time scale, the vehicle may be accelerated through the turbulent flow similarly to the fast-tracking effect (Figure 1a). Under these conditions, we expect the vehicle to be preferentially swept toward the sides of vortices pushing in the direction of motion, and to be accelerated along a fast-tracking trajectory, as shown by the simulated comparison in next section. By taking advantage of beneficial flow structures, the vehicle may achieve a larger average terminal horizontal velocity and travel a longer distance over the same amount of time when compared to other (inefficient) trajectories.

### 2.2. Cellular Flow Fields

For illustration purposes, the control approach presented in this paper is demonstrated for vehicles traversing a two-dimensional cellular flow field with known characteristic parameters. However, the approach can be extended to other flow structures for which fast-tracking results are also available [49,50,51]. Cellular flow is an idealized model of homogeneous turbulent flow that contains a periodic array of eddies described by the vortex length scale [18]. As shown in Figure 2, vortices located in adjacent cells swirl in opposite directions. The cellular flow field is chosen here because it captures essential features of fast-tracking phenomena observed in fully turbulent flows, with some important exceptions described in [21,49,50]. Furthermore, cellular flow represents the best-case scenario for a vehicle in turbulence in the sense that there exist paths for which the flow always provides a tailwind and never a headwind. Finally, by demonstrating the novel fast-tracking control approach in cellular flow, the results may be applicable to a broad range of natural flow phenomena, including but not limited to Langmuir cells in water bodies [58,59] and convective cellular motions in clouds [57].

Given a characteristic flow velocity $U_{0}$, the horizontal and vertical components of the two-dimensional cellular flow velocity (yellow vectors in Figure 2) can be modeled as,
(5a)$$\begin{array}{cc} w_{x} & =U_{0}\sin\left(\frac{\pi x}{L_{w}}\right)\cos\left(\frac{\pi y}{L_{w}}\right) \end{array}$$
(5b)$$\begin{array}{cc} w_{y} & =-U_{0}\cos\left(\frac{\pi x}{L_{w}}\right)\sin\left(\frac{\pi y}{L_{w}}\right) \end{array}$$
respectively, where *x* and *y* are the coordinates in the plane, and $L_{w}$ is a characteristic parameter that represents the distance between two adjacent vortices and is known as the vortex length scale. Together with the $U_{0}$, the vortex length scale, $L_{w}$, determines the vortex time scale,
(6)$$\tau_{w}\equiv \frac{L_{w}}{U_{0}}.$$
which represents the vortex turnover time.

According to the fast-tracking phenomenon, a particle traversing a cellular flow field makes use of the flow structure to travel faster through it when its inertial response time, τ, is approximately equal to the vortex time scale defined in (Equation 6). As illustrated by the simulated blue trajectory in Figure 2, a fast-tracking particle reaches a higher mean settling velocity. Hence, when compared to particles characterized by very different mass and diameter (e.g., red dashed line in Figure 2), a fast-tracking particle travels a much greater (horizontal) distance over the same period of time.

Inspired by these natural phenomenon, this paper develops a feedback control approach devised to allow air vehicles to make use of the eddies to traverse the cellular flow efficiently. By using an implicit model following approach, knowledge of the vortex time scale is used to develop a feedback control law that leverages the fast-tracking effect, irrespective of the vehicle’s mass and size. In the proposed approach, the gravitational force acting on the particle is replaced by a controllable horizontal thrust force (Figure 2) acting on the vehicle by virtue of an onboard propulsion mechanism, such as a propeller or jet engine.

## 3. Problem Formulation and Assumptions

Although the problems of guidance and control in turbulence have been investigated extensively to date [71,72,73,74,75,76,77], previous approaches have focused on attenuating the influence of external wind forces and moments by methods known as disturbance rejection. Besides being applicable only for small disturbances with known and well-posed statistics, such as zero mean and Gaussian characteristics, previous approaches sought to eliminate wind effects, rather than to exploit them as do natural flyers [1,2,19]. As in the extensive literature on trajectory planning for fixed-wing aircraft [44,66,67] and quadcopters [68,69], let the air vehicle be approximated by a point mass and denote its mass by $m_{v}$ and its diameter by $D_{v}$. The point-mass assumption is effective in practice when the vehicle geometry can be ignored in obstacle avoidance problems, and the vehicle size is much smaller than the vortex length scale, or $D_{v}<<L_{w}$. Typically, the size of small unmanned aerial vehicles (UAVs) spans from around 15 cm to 2 m, and the length scale of the energetic turbulent eddies in the atmosphere is about 100 m [62,78,79]. Furthermore, as illustrated in Figure 2, the vehicle propelled by a thrust force must traverse a cellular-flow wind field with vortex length scale, $L_{w}$, and time scale, $\tau_{w}$. For simplicity, the lift force is assumed appropriate for maintaining the vehicle aloft or, alternatively, the vehicle may be assumed neutrally buoyant [21]. Additionally, it is assumed by the same rationale that the effects of the vehicle on the surrounding flow are negligible.

Because the vehicle may encounter different flow fields during its operations and its physical characteristics (mass and diameter) are fixed a priori, its inertial response time, $\tau_{v}$, may not always be approximately equal to $\tau_{w}$. Therefore, in general, the vehicle may not experience the fast-tracking effect. The problem considered in this paper is to develop a feedback control law that modifies the vehicle’s inertial response time in the closed loop, so as to achieve fast tracking by virtue of the controllable thrust forces. It is assumed that the fluid flow dynamic viscosity, μ, and time scale, $\tau_{w}$, are either known a priori or estimated from wind measurements online, for example, using the sparse identification of nonlinear dynamics (SINDy) [55,80]. The vehicle physical parameters are lumped into a constant vector, $\theta ={\left[m_{v}\;D_{v}\right]}^{T}$, and the onboard propulsion produces a constant horizontal thrust, $T_{x}=m_{v}a_{x}$, as well as acceleration-based control inputs, $u={\left[u_{x}\;u_{y}\right]}^{T}$. The vehicle acceleration produced by the constant horizontal thrust is denoted by $a={\left[a_{x}\;0\right]}^{T}$. Then, from transport theory [50], the two-dimensional vehicle dynamics subject to a cellular flow can be modeled by a linear parameter-dependent system,
(7)$$\dot{x}\left(t\right)=A\left(\theta ,\mu \right)x\left(t\right)+B\left(\theta ,\mu \right)u\left(t\right)+L\left(\theta ,\mu \right)w\left(t\right)+a,\;x\left(t_{0}\right)=x_{0},$$
where the state vector, $x={\left[v_{x}\;v_{y}\right]}^{T}$, consists of the *x*- and *y*-components of the vehicle velocity in inertial frame, and the wind flow velocity vector, denoted by $w={\left[w_{x}\;w_{y}\right]}^{T}$, is assumed known from onboard measurements. The initial conditions, $x_{0}$, are known from the vehicle. The state-space matrices are given by,
(8)$$A\left(\theta ,\mu \right)=\left[\begin{array}{cc} -\frac{3\pi D_{v}\mu }{m_{v}} & 0\\ 0 & -\frac{3\pi D_{v}\mu }{m_{v}} \end{array}\right];\;B\left(\theta ,\mu \right)=I_{2\times 2};\;L\left(\theta ,\mu \right)=\left[\begin{array}{cc} \frac{3\pi D_{v}\mu }{m_{v}} & 0\\ 0 & \frac{3\pi D_{v}\mu }{m_{v}} \end{array}\right]$$
where it is assumed that the vehicle is subject to a linear Stokes drag force [81]. This assumption is justified when the flow is incompressible and $D_{v}<<L_{w}$, and holds approximately for air vehicles such as fixed-wing aircraft and rotorcraft in high Reynolds number regimes under certain conditions [20,82,83,84]. A state-feedback controller is developed, assuming that the vehicle state is fully observable and estimated with zero error for simplicity [85]. Furthermore, a constant horizontal thrust is provided to obey (Equation 2), such that
(9)$$T_{x}=m_{v}a_{x}=3\pi \mu D_{v}\tau_{v}a_{x}$$

This paper seeks to develop a feedback control system inspired by the natural transport phenomena described in Section 2.1, such that, in the closed loop, the vehicle behaves like a particle undergoing the fast-tracking effect. The desired automatic feedback control law must provide the vehicle inputs u(t), in (Equation 7), continuously over time, so as to exploit the energy and organized structure of the eddies in the cellular flow field. This novel control approach is demonstrated by solving the following benchmark control problems:

(1) Minimum-energy problem: determine control inputs, u(t), so as to reach and maintain a desired steady-state velocity through the cellular flow with minimum control effort.

(2) Minimum-time problem: determine control inputs, u(t), so as to travel a desired distance in the horizontal direction through the cellular flow in minimum time.

The new FTC control approach is derived using implicit model following in the next section. Subsequently, its performance is demonstrated in Section 5 and Section 6, and compared to two classic optimal solutions obtained via linear quadratic regulation and bang-bang control, respectively. Although the approach is demonstrated on the simplified air vehicle model in (Equation 7), the methods proposed in this paper can be easily extended to more detailed vehicle dynamic models, provided they too may be approximated by linear parameter dependent systems.

## 4. FTC Control Design via Implicit Model Following (IMF)

The FTC feedback control design is developed by specifying an implicit model based on the ideal response of an efficient fast-tracking particle in the loop. While most of the existing control methods seek to compensate for or reject wind effects, the FTC control approach seeks to make use of organized cellular flow structures in order to benefit from them in terms of speed and energy consumption. Because the real geometry and location of the eddies are unknown to the vehicle, they may not be utilized for trajectory optimization. Rather, in an effort to mimic natural transport phenomena, the dynamic model of an ideal fast-tracking particle is first obtained from the known parameters of the cellular flow ($L_{w}$ and $\tau_{w}$), as described in Section 2.2. Subsequently, a state-feedback control law is obtained using implicit model following (IMF) [86], such that the closed-loop vehicle dynamics may follow the ideal fast-tracking particle model as closely as possible.

The IMF approach, originally proposed in [86], leverages an implicit dynamic model to obtain a control system that conforms to an ideal behavior. Once an ideal dynamic model with state, $x_{m}\in R^{n}$, is formulated, the IMF control law is obtained by minimizing the error between the time derivatives of the vehicle state, $x\in R^{n}$, and those of the model, i.e.:(10)$$J=\frac{1}{2}\int_{t_{0}}^{t_{f}}\;{\left[\dot{x}\left(t\right)-{\dot{x}}_{m}\left(t\right)\right]}^{T}Q_{m}\left[\dot{x}\left(t\right)-{\dot{x}}_{m}\left(t\right)\right]\;dt,$$
where the subscript *m* refers to “model”. The positive definite weighting matrix $Q_{m}\in R^{n\times n}$ can be utilized to specify a desired trade-off between state variables, for example, in order to account for states’ range and units. The IMF equations and control law are derived in Section 4.2, based on the fast-tracking particle model presented in the next section.

### 4.1. Ideal Fast-Tracking Particle Model

As a first step, let us reinterpret the gravitational force acting on an inertial particle as a constant thrust in the horizontal direction, $T_{x_{m}}$. With a simple coordinate transformation, assuming the air flow is incompressible and the particle diameter, $D_{m}$, is significantly smaller than the vortex length scale, $L_{w}$, the two-dimensional governing equation for a spherical inertial particle of mass *m* subject to a drag force $F_{d}$ and traveling in a cellular flow field is
(11)$$m{\dot{x}}_{m}=F_{d}+T_{m}$$
where $x_{m}={\left[v_{x_{m}}\;v_{y_{m}}\right]}^{T}$ contains the particle velocity in inertial frame, and $T_{m}={\left[T_{x_{m}}\;0\right]}^{T}$ denotes constant external thrust. In this ideal model, the inertial particle is subject to a linear Stokes drag force,
(12)$$F_{d}=-3\pi D_{m}\mu \left(x_{m}-w_{m}\right)$$
where $w_{m}={\left[w_{x_{m}}\;w_{y_{m}}\right]}^{T}$ is the flow velocity in inertial frame [81,87]. As explained in Section 3, the above assumption holds, once again, because it can be assumed that the flow is incompressible and $D_{m}<<L_{w}$ [20,82,83,84]. In spite of all these assumptions, the ideal particle model presented in this section can well explain many natural particle transport phenomena, such as water droplets settling in air turbulence [88] and soot formation in turbulent flames [89].

When the inertial response time of the ideal particle, $\tau_{m}$, is approximately equal to $\tau_{w}$, the particle exhibits the fast-tracking effect and naturally follows the most efficient trajectories inside the cellular flow. From the equations of the particle’s inertial response time (Equation 2) and the particle model (Equation 11), the particle dynamics can be expressed as a linear parameter-dependent system,
(13)$${\dot{x}}_{m}\left(t\right)=A_{m}\left(\tau_{m}\right)x_{m}\left(t\right)+L_{m}\left(\tau_{m}\right)w_{m}\left(t\right)+a_{m},\;x_{m}\left(t_{0}\right)=x_{0_{m}},$$
where $a_{m}=\frac{1}{m}T_{m}={\left[a_{x_{m}}\;0\right]}^{T}$ is the particle acceleration produced by the constant horizontal thrust, $x_{0_{m}}$ are the particle’s initial conditions, and model state-space matrices, $A_{m}$ and $L_{m}$, depend only on the ideal particle’s inertial response time, $\tau_{m}$, which is chosen to match the vortex time scale. The state-space matrices are given by
(14)$$A_{m}\left(\tau_{m}\right)=\left[\begin{array}{cc} -\frac{1}{\tau_{m}} & 0\\ 0 & -\frac{1}{\tau_{m}} \end{array}\right];\;L_{m}\left(\tau_{m}\right)=\left[\begin{array}{cc} \frac{1}{\tau_{m}} & 0\\ 0 & \frac{1}{\tau_{m}} \end{array}\right]$$

### 4.2. Fast-Tracking Controller (FTC) Design

Unlike the ideal particle described in the previous section, in general, the vehicle has an inertial response time that is not approximately equal to the vortex time scale. Therefore, a feedback control law can be derived to change the vehicle response in the closed loop and make it follow the behavior of the ideal particle model, which is implicit in the law itself. Choose $w=w_{m}$, and construct a quadratic cost function in the form (Equation 10),
(15)$$\begin{array}{cc} J & =\frac{1}{2}\int_{t_{0}}^{t_{f}}\;{\left[\dot{x}\left(t\right)-{\dot{x}}_{m}\left(t\right)\right]}^{T}Q_{m}\left[\dot{x}\left(t\right)-{\dot{x}}_{m}\left(t\right)\right]\;dt\\ \; & =\frac{1}{2}\int_{t_{0}}^{t_{f}}\;\left[x^{T}\left(t\right)Q\;x\left(t\right)+2x^{T}\left(t\right)M\;\tilde{u}\left(t\right)+{\tilde{u}}^{T}\left(t\right)R\;\tilde{u}\left(t\right)\right]\;dt, \end{array}$$
where $\tilde{u}=u+\left(\frac{1}{\tau_{v}}-\frac{1}{\tau_{m}}\right)w+a-a_{m}$, $Q_{m}=I_{2\times 2}$, and the weighing matrices are designed as follows,
(16)$$\begin{array}{cc} \; & Q={\left(A-A_{m}\right)}^{T}Q_{m}\left(A-A_{m}\right)\\ \; & M={\left(A-A_{m}\right)}^{T}Q_{m}B\\ \; & R=B^{T}Q_{m}B \end{array}$$
in order to minimize (Equation 15).

For the vehicle model shown in Section 3, perfect model following can be achieved (with zero state error) because the following perfect-model-following criterion is satisfied,
(17)$$\left({\mathrm{BB}}^{L}-I_{n}\right)\left(A-A_{m}\right)=0,$$
where $B^{L}={\left(B^{T}B\right)}^{-1}B^{T}$ is the left pseudo-inverse [86]. When the FTC approach is extended to other vehicle dynamics, the above criterion may not be always satisfied and, hence, the optimal IMF control law described in [90] can be adopted to minimize the model-following error. In particular, letting $t_{f}$ approach infinity in (Equation 15), the optimal control law can be obtained in terms of a steady-state gain matrix C(0) such that,
(18)$$\tilde{u}\left(t\right)=-C\left(0\right)x\left(t\right)=-R^{-1}\left[B^{T}S\left(0\right)+M^{T}\right]x\left(t\right)$$
where S(0) is the solution of the algebraic Riccati equation (ARE),
(19)$$\begin{array}{cc} \left[S\left(0\right)B+M\right]R^{-1}[B^{T}S\left(0\right) & +M^{T}]-A^{T}S\left(0\right)-S\left(0\right)A-Q=0 \end{array}$$

In this case, the Riccati matrix solution that guarantees closed-loop asymptotic stability is $S\left(0\right)=0_{2\times 2}$, and the corresponding steady-state gain matrix is
(20)$$C\left(0\right)=R^{-1}\left[B^{T}S\left(0\right)+M^{T}\right]=\left(\frac{1}{\tau_{m}}-\frac{1}{\tau_{v}}\right)I_{2\times 2}$$

Hence, under the aforementioned assumptions, the FTC IMF feedback control law is
(21)$$\tilde{u}\left(t\right)=-C\left(0\right)x\left(t\right)=-\left(\frac{1}{\tau_{m}}-\frac{1}{\tau_{v}}\right)x\left(t\right)$$
which is implemented as the following acceleration-based control inputs onboard the air vehicle,
(22)$$u\left(t\right)=\tilde{u}\left(t\right)-\left(\frac{1}{\tau_{v}}-\frac{1}{\tau_{m}}\right)w\left(t\right)-a+a_{m}=\left(\frac{1}{\tau_{v}}-\frac{1}{\tau_{m}}\right)\left[x\left(t\right)-w\left(t\right)\right]-a+a_{m}$$

It can be seen that the FTC control law requires only online measurements of the vehicle state, x, and of the local wind flow field, w, both of which may be estimated with excellent accuracy onboard many vehicles. As a result, without knowledge of the global wind profile (or of the precise eddies’ positions and geometries), the FTC-controlled vehicle finds the most efficient trajectories achievable by the vehicle based on its dynamic constraints (Equation 7).

## 5. Minimum-Energy Solutions and Results

The utilization of aerial vehicles, especially small ones, is often limited by the allowed on-board battery capacity and duration of flight, particularly during rapid and aggressive maneuvers in extreme windy conditions [21,91,92]. Therefore, energy consumption has become an essential performance metric for small UAV control design. In this benchmark minimum-energy control problem, a feedback control law is desired to make the vehicle reach and maintain a desired steady-state velocity, $x_{r}={\left[v_{x_{r}}\;v_{y_{r}}\right]}^{T}$, through the cellular flow with minimum control effort. For comparison, the desired steady-state velocity $x_{r}$ is chosen to be the mean settling velocity of the ideal fast-tracking particle. The classic optimal solution to this problem obtained by linear quadratic regulation will be proposed in Section 5.1. Then, the performance of FTC will be demonstrated by comparing it with the classic linear quadratic regulator (LQR) in two different case studies. In practice, the vehicle’s inertial response time, $\tau_{v}$, is not approximately equal to that of the ideal fast-tracking particle, $\tau_{m}$. Therefore, FTC and LQR are implemented and compared for the control of an air vehicle with $\tau_{v}$ greater than $\tau_{m}$ in Section 5.2, and an air vehicle with $\tau_{v}$ smaller than $\tau_{m}$ in Section 5.3, respectively. In the end, based on multiple numerical simulations, the dependence of the minimum-energy simulation results on the choice of $\tau_{v}$ with respect to $\tau_{m}$ is discussed.

### 5.1. Comparison with Linear Quadratic Regulator (LQR)

LQR is an optimal solution to the control problem of making the vehicle reach and maintain a desired steady state velocity. The flow disturbances perpendicular to the desired steady state velocity are compensated by state feedback control. We first rewrite the two-dimensional vehicle dynamics (Equation 7) in the standard state-space form,
(23)$$\dot{x}\left(t\right)=\mathrm{Ax}\left(t\right)+B\bar{u}\left(t\right),$$
where $\bar{u}=u+a+\frac{1}{\tau_{v}}w$. Whereas, in the previous section, the cost function (Equation 15) was an integral of the difference between the state derivatives of the air vehicle and the fast-tracking particle, we construct the cost function as a different integral here that penalizes both state excursions and control effort for this particular problem,
(24)$$J=\frac{1}{2}\int_{t_{0}}^{t_{f}}\;\left[x^{T}\left(t\right)Q\;x\left(t\right)+{\bar{u}}^{T}\left(t\right)R\;\bar{u}\left(t\right)\right]\;dt$$
where $Q=R=I_{2\times 2}$. With perfect knowledge of the vehicle state, the desired feedback control law, $\bar{u}$, can be expressed in terms of the steady-state gain matrix C(0) as $t_{f}$ approaches infinity such that,
(25)$$\begin{array}{cc} \bar{u}\left(t\right) & =-C\left(0\right)x\left(t\right)+K_{r}x_{r}\\ \; & =-R^{-1}B^{T}S\left(0\right)x\left(t\right)+K_{r}x_{r} \end{array}$$
where S(0) is a solution to the algebraic Riccati equation (ARE),
(26)$$A^{T}S\left(0\right)+S\left(0\right)A-S\left(0\right){\mathrm{BR}}^{-1}B^{T}S\left(0\right)+Q=0$$
and $K_{r}$ can be chosen to track the reference $x_{r}$ with zero steady-state error [93],
(27)$$K_{r}=-{\left\{{\left[A-\mathrm{BC}\left(0\right)\right]}^{-1}B\right\}}^{-1}$$

In this case, the Riccati matrix solution that guarantees closed-loop asymptotic stability is
(28)$$S\left(0\right)=-\left(-\frac{1}{\tau_{v}}+\sqrt{\frac{1}{\tau_{v}^{2}}+1}\;\right)\;I_{2\times 2}$$

Combining (Equation 25), (Equation 27), and (Equation 28), we can obtain the following acceleration-based LQR feedback control law,
(29)$$u\left(t\right)=\bar{u}\left(t\right)-a-\frac{1}{\tau_{v}}w\left(t\right)=-\left(-\frac{1}{\tau_{v}}+\sqrt{\frac{1}{\tau_{v}^{2}}+1}\;\right)\;x\left(t\right)+\sqrt{\frac{1}{\tau_{v}^{2}}+1}\;x_{r}-a-\frac{1}{\tau_{v}}w\left(t\right)$$

### 5.2. FTC Minimum-Energy Case Study 1

In this case study, FTC and LQR are implemented and compared for the control of a thrust-driven vehicle with $\tau_{v}>\tau_{m}$. In the simulation, the controlled vehicle with $\tau_{v}$ = 0.21 s traverses the cellular flow with $\tau_{w}$ = 0.15 s for 20 s. Additionally, another purely forward-thrust-driven vehicle with the same $\tau_{v}$ as the controlled vehicles and an ideal fast-tracking particle with $\tau_{m}=\tau_{w}$ are simulated for comparison. As previously explained in Section 3, to keep the control cost comparable among different control designs and case studies, the horizontal thrust in (Equation 9) remains unchanged when the vehicle’s inertial response time changes in different simulations. Therefore, $\tau_{v}a_{x}$ stays the same at 1.5 m/s here. In this FTC minimum-energy simulation, the fast-tracking particle, the vehicle purely driven by forward thrust, and FTC-controlled and LQR-controlled vehicles all start from the origin, and their initial velocities are assumed to be zero. FTC aims to make the air vehicle follow the ideal response of the fast-tracking particle. The control objective is to make the air vehicle reach and maintain a desired steady-state velocity, $x_{r}={\left[v_{x_{r}}\;v_{y_{r}}\right]}^{T}$, where $v_{x_{r}}$ = 15.41 m/s and $v_{y_{r}}$ = 0 m/s, through the cellular flow with minimum control effort.

Trajectories of the fast-tracking particle with $\tau_{m}=0.15$ s, and purely thrust-driven, FTC-controlled, and LQR-controlled vehicles with $\tau_{v}=0.21$ s are compared in Figure 3. They all start from the origin at the same time, go through a transitional period in the cellular flow, and travel horizontally in the end. Nevertheless, the LQR-controlled vehicle travels horizontally without any oscillations, because its vertical velocity maintains zero all the time to meet the control objective. The trajectory of the FTC-controlled air vehicle identically overlaps that of the fast-tracking particle, which demonstrates that the FTC-controlled air vehicle can perfectly follow the fast-tracking particle with zero tracking error. They both go through a short transitional period at first, and are then swept and, thus, readily accelerated into the downward sweeping sides of eddies. However, the air vehicle purely driven by constant horizontal forward thrust deviates a little from the desired horizontal direction at first, and takes a longer transitional time to adapt to the cellular flow conditions.

In Figure 4, the velocity time histories of the fast-tracking particle and the purely thrust-driven, FTC-controlled, and LQR-controlled vehicles are compared. The FTC-controlled and LQR-controlled air vehicles both meet the control objective of reaching and maintaining a desired steady-state velocity, $x_{r}={\left[v_{x_{r}}\;v_{y_{r}}\right]}^{T}$, where $v_{x_{r}}=15.41$ m/s and $v_{y_{r}}=0$ m/s. With the flow disturbance compensated, the LQR-controlled air vehicle achieves and maintains the desired steady-state velocity perfectly with zero tracking error, while the horizontal velocity component, $v_{x}$, of the FTC-controlled air vehicle oscillates a little around the desired horizontal steady-state velocity, $v_{x_{r}}$, due to the periodically changing flow conditions. At the expense of sacrificing the vehicle’s riding comfort, the FTC-controlled vehicle exploits beneficial flow structures and harvests energy from cellular flows. Additionally, the LQR-controlled air vehicle reaches $v_{x_{r}}$ faster than the FTC-controlled one. However, the vehicle purely driven by forward thrust takes a longer transitional time to adapt to the fluctuating flow conditions. According to Figure 4, the average horizontal velocities of the LQR-controlled and FTC-controlled air vehicles over the entire period of simulation are both significantly greater than the vehicle purely driven by forward thrust. Consequently, both LQR-controlled and FTC-controlled vehicles travel a longer distance horizontally than the purely thrust-driven vehicle within the same period of time.

In this benchmark minimum-energy problem, the objective is to use minimum control effort. Therefore, the quadratic control usage, given by $u^{T}u=u_{x}^{2}+u_{y}^{2}$, of FTC is compared to that of LQR in Figure 5. The total control effort of a controller is commonly quantified by the integral quadratic control usage *C*, which takes the form,
(30)$$C=\int_{t_{0}}^{t_{f}}\;u^{T}\left(t\right)u\left(t\right)\;dt.$$

In this case study, over the same period from time $t_{0}=0$ s to time $t_{f}=20$ s, the integral quadratic control usage of FTC is $C_{f}=6.71\times 10^{3}$ m^2^/s^3^, while that of LQR is $C_{l}=1.33\times 10^{5}$ m^2^/s^3^. LQR has a much larger control cost than FTC.

To represent the difference of inertial response time between the controlled air vehicle and the ideal fast-tracking particle, the ratio of inertial response time is defined as
(31)$$\eta_{\tau }=\frac{\tau_{v}}{\tau_{m}}$$

Similarly, to quantify the relative control savings of FTC compared to LQR, the ratio of total control cost is defined as
(32)$$\eta_{c}=\frac{C_{l}}{C_{f}}$$

Therefore, we can obtain that, in case study 1, the ratio of inertial response time is $\eta_{\tau }=1.40$, and the ratio of total control cost is $\eta_{c}=19.83$.

### 5.3. FTC Minimum-Energy Case Study 2

In this case study, FTC and LQR are implemented for the control of a thrust-driven vehicle with $\tau_{v}=0.075$ s $<\tau_{m}$. The cellular flow parameters, control objective, and simulation conditions are all the same as in case study 1. Another purely forward-thrust-driven vehicle with $\tau_{v}=0.075$ s and an ideal particle are also simulated for comparison. To keep the control cost comparable among different cases, the horizontal thrust $T_{x}$, or $\tau_{v}a_{x}$ equivalently, remains the same as in case study 1. Trajectories of the ideal particle and purely thrust-driven, FTC-controlled, and LQR-controlled vehicles are compared in Figure 6. Similarly to case study 1, the FTC-controlled vehicle can perfectly follow the ideal particle with zero tracking error.

In Figure 7, the horizontal velocity and quadratic control usage of the two controlled vehicles are compared. They both meet the velocity tracking control objective. Additionally, the LQR-controlled vehicle reaches $v_{x_{r}}$ faster than the FTC-controlled one, and its settling time is less than half of the settling time in case study 1. The purely forward-thrust-driven vehicle behaves similarly in each case study with a much longer settling time and smaller average settling velocity compared to the controlled vehicles. Over the same period of simulation, the integral quadratic control usage of FTC is $C_{f}=8.22\times 10^{4}$ m^2^/s^3^, while that of LQR is $C_{l}=1.04\times 10^{6}$ m^2^/s^3^. Therefore, LQR has a larger control cost than FTC. Accordingly, the ratio of inertial response time is $\eta_{\tau }=0.5$, and the ratio of total control cost is $\eta_{c}=12.69$ in this case.

Through multiple numerical simulations with the ratio of inertial response time $\eta_{\tau }$ ranging from 0.01 to 500, we find that there is a trade-off between the control savings of FTC and the difference of inertial response time between the controlled vehicle and the fast-tracking particle. The larger the difference of inertial response time is, the more FTC control effort will be cost to make the vehicle follow the ideal response of the fast-tracking particle. In Figure 8, the polynomial fits to the circular and cross data points illustrate how the relative total control cost, $\eta_{c}=C_{l}/C_{f}$, changes as a function of the normalized inertial response time of the vehicle, $\eta_{\tau }=\tau_{v}/\tau_{m}$. The relative total control cost, $\eta_{c}$, approaches infinity when $\eta_{\tau }=1$, because the total control cost of FTC is zero when the inertial response time of the vehicle is equal to that of the fast-tracking particle. On the left side of the vertical asymptote at $\eta_{\tau }=1$, all the red cross markers correspond to simulations where $\tau_{v}<\tau_{m}$, and the yellow one represents case study 2. In this area, the relative total control cost, $\eta_{c}$, is enhanced as the normalized inertial response time, $\eta_{\tau }$, grows. On the right side of the vertical asymptote at $\eta_{\tau }=1$, all the blue circular markers correspond to simulations where $\tau_{v}>\tau_{m}$, and the yellow one represents case study 1. Conversely, in this area, $\eta_{c}$ decreases as $\eta_{\tau }$ grows. Moreover, there exists two horizontal asymptotes for the polynomial fits: $\eta_{c}$ approaches $1.50\times 10^{-3}$ as $\eta_{\tau }$ increases, and approaches 3.19 as $\eta_{\tau }$ decreases. In addition, the logarithms of the two ratios $\eta_{c}$ and $\eta_{\tau }$ are approximately first-order linearly dependent with each other when $\eta_{\tau }$ is roughly within the range of 2 to 10. All the data points located in the grey shaded area above the horizontal dashed line at $\eta_{c}=1$ correspond to simulations in which the FTC-controlled air vehicle achieves the control objective with less control effort than the LQR-controlled one, while those below the dashed line at $\eta_{c}=1$ correspond to simulations where FTC costs more control effort than LQR. The critical point where $\eta_{c}=1$ locates approximately at $\eta_{\tau }^{*}\approx 2.80$. Therefore, below this critical inertial response time ratio, the FTC-controlled vehicle uses less energy to traverse the cellular flow compared with the LQR-controlled one.

## 6. Minimum-Time Solutions and Results

As UAVs have been widely used, there is an increasing need to extend the range and endurance of UAV flight in many applications with extremely strict time limitations, including autonomous medical delivery and emergency response [94,95]. In addition to energy consumption, the cost of time has become another essential factor to consider for UAV path planning and control. In this benchmark minimum-time control problem, we aim to find an optimal control law for the air vehicle with $\tau_{v}$, such that the controlled vehicle can travel a desired distance in the horizontal direction, $x_{d}$, through the cellular flow with $\tau_{w}=\tau_{m}$ within a minimum time. The optimal solution to this problem, a bang-bang controller (BBC), will be proposed in Section 6.1. Subsequently, the performance of FTC will be demonstrated by comparing it with the optimal BBC solution in Section 6.2. In the end, based on multiple numerical simulations, the dependence of the minimum-time simulation results on the choice of $\tau_{v}$ with respect to $\tau_{m}$ is discussed.

### 6.1. Comparison with Bang-Bang Controller (BBC)

In still fluid, bang-bang control, which requires full use of available control effort, is the optimal solution to the minimum-time problem when the control is bounded [86]. In this paper, since the horizontal and vertical motions of the vehicle are decoupled, we first consider the horizontal vehicle dynamics and ignore the flow disturbance for simplicity. Additionally, the vertical control input, $u_{y}$, is assumed to be zero. Assuming that the vehicle state is represented by $x={\left[x\;v_{x}\right]}^{T}$, the horizontal vehicle dynamics can be expressed as
(33)$$\left[\begin{array}{c} \dot{x}\\ {\dot{v}}_{x} \end{array}\right]=\left[\begin{array}{cc} 0 & 1\\ 0 & -1/\tau \end{array}\right]\left[\begin{array}{c} x\\ v_{x} \end{array}\right]+\left[\begin{array}{c} 0\\ 1 \end{array}\right]{\hat{u}}_{x}$$
where ${\hat{u}}_{x}=u_{x}+a_{x}$, and the horizontal control input is assumed to be bounded,
(34)$$u_{x}=u_{x}\left(t\right),\;-u_{0}⩽u_{x}⩽u_{0}.$$

To minimize the time to reach the desired horizontal position, $x_{d}$, the cost function can be constructed as,
(35)$$J=\int_{t_{0}}^{t_{f}}\;1\;dt$$
subject to the horizontal vehicle dynamics in (Equation 33) and boundary conditions,
(36)$$x\left(t_{0}\right)=v_{x}\left(t_{0}\right)=0,\;x\left(t_{f}\right)=x_{d},\;v_{x}\left(t_{f}\right)=v_{x_{r}},$$
where $v_{x_{r}}$ is the horizontal steady-state velocity achieved by the ideal fast-tracking particle. In general, the optimal control history for this type of minimum-time problem takes the form [96],
(37)$$u_{x}\left(t\right)=\left\{\begin{array}{cc} u_{0} & t_{0}⩽t⩽t_{w}\\ -u_{0} & t_{w}<t⩽t_{f} \end{array}\right.$$
where $t_{w}$ is the switching time, and $t_{f}$ is the terminal time. The control switches at time $t_{w}$ and only takes the boundary value. Then, the switching time $t_{w}$ and the terminal time $t_{f}$ can be obtained by solving the constrained optimization problem formulated above.

### 6.2. FTC Minimum-Time Case Study 3

In this case study, FTC and BBC are implemented and compared for the control of a thrust-driven vehicle with $\tau_{v}=0.225$ s traversing a cellular flow with $\tau_{w}=0.15$ s. In the simulation, the FTC-controlled and BBC-controlled vehicles both start from the origin with zero initial velocity, and the control is assumed to be bounded: $-u_{0}⩽u_{x}⩽u_{0}$, where $u_{0}=8$ m/s^2^. The control objective is to make the air vehicle travel a desired distance in the horizontal direction, $x_{d}=15$ m, through the cellular flow using minimum time. The FTC-controlled air vehicle achieves this objective by following the ideal response of a fast-tracking particle with $\tau_{m}=\tau_{w}$.

By solving the constrained optimization problem with the flow disturbance ignored in Section 6.1, we find that if the control switches at time $t_{w}=6.54$ s, the BBC-controlled air vehicle can reach the desired horizontal position at time $t_{f}=6.55$ s in still fluid. Although we neglect the flow disturbance for simplicity when deriving the optimal bang-bang control law, the flow effects on the air vehicle are still considered in the simulation. As shown in Figure 9, both the FTC-controlled and BBC-controlled air vehicles start at the same time from the origin, go through a transitional period in the cellular flow, and travel horizontally in the end. However, compared to the FTC-controlled one, the BBC-controlled air vehicle significantly deviates from the desired horizontal direction at first, and takes a longer transitional time to reach an approximate equilibrium state.

Figure 10 shows the horizontal velocity and position time histories of the FTC-controlled and BBC-controlled air vehicles. The time-average horizontal velocity of the BBC-controlled air vehicle is smaller than that of the FTC-controlled one. Consequently, in the simulation, the FTC-controlled air vehicle reaches the desired position, $x_{d}=15$ m, at time $t_{a}=2.47$ s, while the BBC-controlled one reaches the desired position at time $t_{b}=5.09$ s. Even though the bang-bang control law is the optimal solution to the benchmark minimum-time problem for bounded control inputs, the FTC-controlled air vehicle achieves the control objective of reaching a desired horizontal position through the cellular flow using less time compared to the BBC-controlled one.

The comparison of the quadratic control usage, $u^{T}u=u_{x}^{2}+u_{y}^{2}$, of FTC and BBC is shown in Figure 11. Given that the FTC-controlled and BBC-controlled vehicles travel the same desired horizontal distance, the integral quadratic control usage of FTC over the entire period from time $t_{0}=0$ s to time $t_{a}=2.47$ s is $C_{f}=266.73$ m^2^/s^3^, while the integral quadratic control usage of BBC over the entire period from time $t_{0}=0$ s to time $t_{b}=5.09$ s is $C_{b}=325.89$ m^2^/s^3^. The comparison of FTC and BBC performance is shown in Table 1. Compared to the BBC-controlled one, the FTC-controlled air vehicle achieves the control objective of reaching a desired horizontal position through the cellular flow not only within a smaller period of time, but also with less control effort.

In this case study, the ratio of inertial response time is $\eta_{\tau }=1.5$. Similarly, to quantify the relative time savings of FTC compared to BBC, the ratio of total time is defined as
(38)$$\eta_{t}=\frac{t_{b}}{t_{a}}=2.06$$
where $t_{a}$ denotes the total time spent by the FTC-controlled vehicle to reach the desired horizontal position, and $t_{b}$ denotes that of the BBC-controlled vehicle. To analyze how the ratio of total time, $\eta_{t}$, changes as a function of the normalized inertial response time of the vehicle, $\eta_{\tau }$, we perform multiple numerical simulations with $\eta_{\tau }$ ranging from 0.1 to 10. As shown in Figure 12, all the data points located in the grey shaded area above the horizontal dashed line at $\eta_{t}=1$ correspond to simulations in which the FTC-controlled vehicle spends less time traversing the flow compared with the BBC-controlled one. The critical point where $\eta_{t}=1$ locates approximately at $\eta_{\tau }^{*}\approx 6$. The yellow circular marker represents the simulation chosen as the case study in this section. As $\eta_{\tau }$ increases, $\eta_{t}$ grows at first, reaches a maximum, and then starts to decrease when $\eta_{\tau }\approx 3.5$. Moreover, there exists a horizontal asymptote for the polynomial fit: $\eta_{t}$ approaches 0.8 as $\eta_{\tau }$ keeps increasing.

## 7. Summary and Conclusions

This paper presents a novel and general control design approach, referred to as a fast-tracking controller or FTC in short, applicable to air vehicles flying through cellular-flow-like turbulent fields. The new design philosophy presented in this paper allows the vehicle, with a feedback controller in the loop, to behave like an ideal fast-tracking particle by means of implicit model following. The simulation results show that, indeed, FTC control allows the vehicle to take advantage of beneficial tailwinds by means of onboard propulsion and local wind measurements, ultimately reducing the travel time and energy consumption required to traverse the cellular flow. The energy-harvesting potential of FTC is demonstrated by considering two benchmark control problems: the minimum-energy problem and minimum-time problem. The comparison with classic optimal solutions obtained via LQR and BBC control theory shows that by following the ideal response of the fast-tracking particle, the FTC-controlled air vehicle achieves a larger average horizontal velocity than the purely thrust-driven one. Furthermore, the FTC-controlled vehicle can reach and maintain a desired steady-state velocity through the cellular flow with less control effort than the LQR-controlled vehicle, and reach a desired horizontal position faster than the BBC-controlled vehicle. Finally, the FTC design presented in this paper is also potentially applicable to underwater vehicles in Langmuir-type water cells.

## Figures and Tables

**Figure 1 biomimetics-07-00192-f001:**
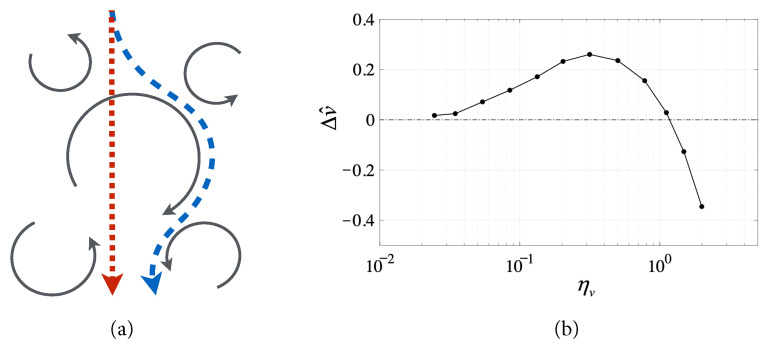
(**a**) Particles are swept and, thus, readily accelerated into the downward sweeping sides of eddies (blue trajectory) rather than falling straight through turbulence (red trajectory). As a result, (**b**) water droplets in turbulent air experience increased settling velocity ($△\hat{v}$) when the particle settling parameter ($\eta_{v}$) is of order one (experimental data taken from [49]).

**Figure 2 biomimetics-07-00192-f002:**
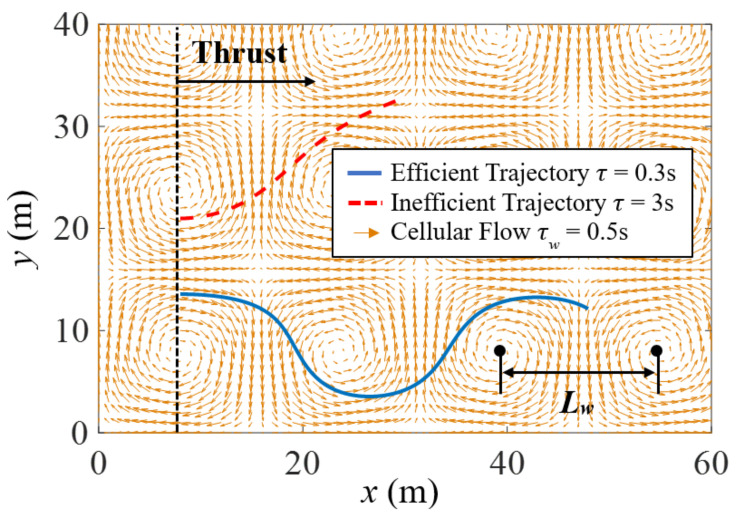
The trajectories of two particles with different inertial response times (τ) are compared by allowing them to travel for the same amount of time through a cellular flow field with vortex time scale ($\tau_{w}$) after they are both released at the dashed black line (see [70] for animation).

**Figure 3 biomimetics-07-00192-f003:**
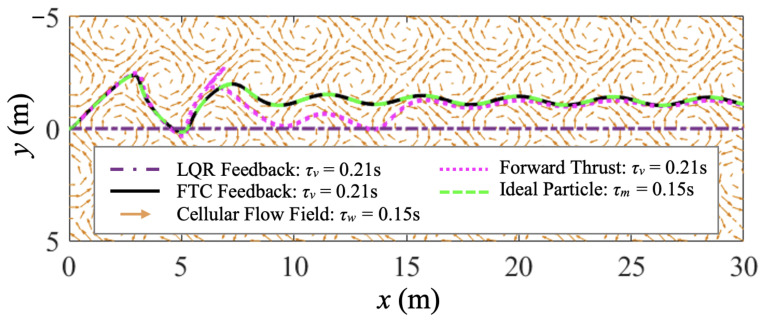
Case study 1: the trajectory comparison of the ideal fast-tracking particle with $\tau_{m}=0.15$ s, and purely thrust-driven, FTC-controlled, and LQR-controlled vehicles with $\tau_{v}=0.21$ s traversing a cellular flow with $\tau_{w}=0.15$ s demonstrates that the FTC-controlled vehicle can perfectly follow the fast-tracking particle with zero tracking error.

**Figure 4 biomimetics-07-00192-f004:**
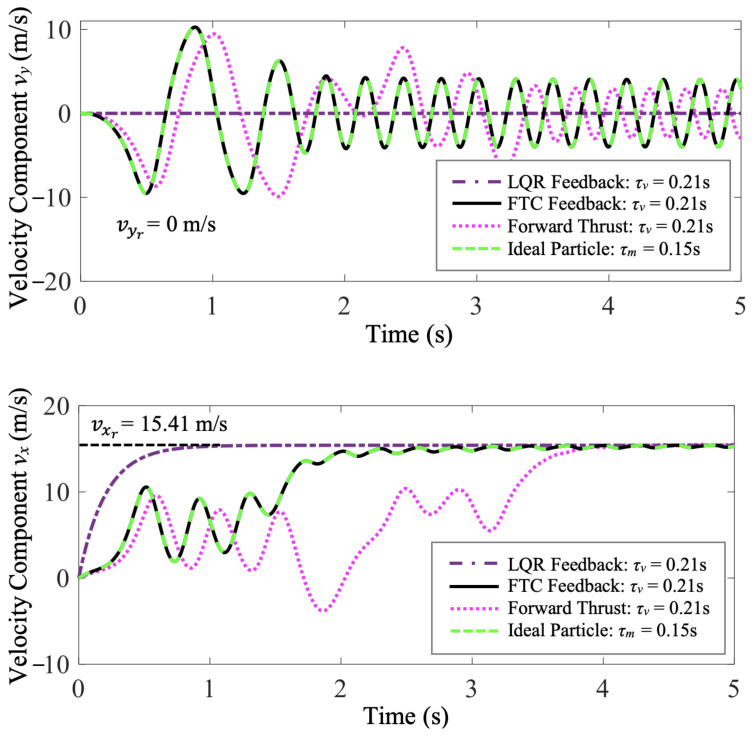
Case study 1: the comparison of the velocity time histories of FTC-controlled and LQR-controlled vehicles shows that they both achieve and maintain the desired steady-state velocities ($v_{x_{r}}$ and $v_{y_{r}}$).

**Figure 5 biomimetics-07-00192-f005:**
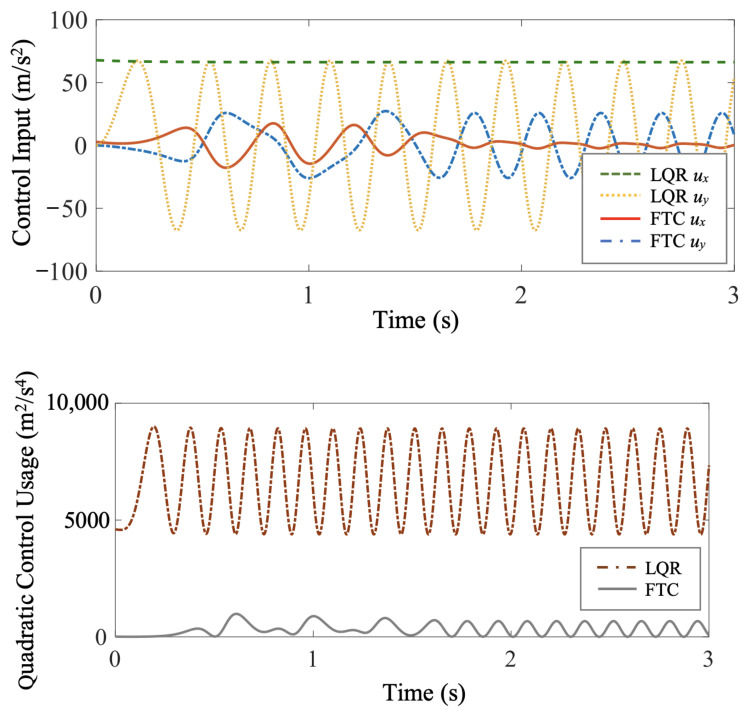
Case study 1: the comparison of the FTC and LQR control cost shows that the FTC-controlled vehicle meets the control objective with much less control effort than the LQR-controlled one.

**Figure 6 biomimetics-07-00192-f006:**
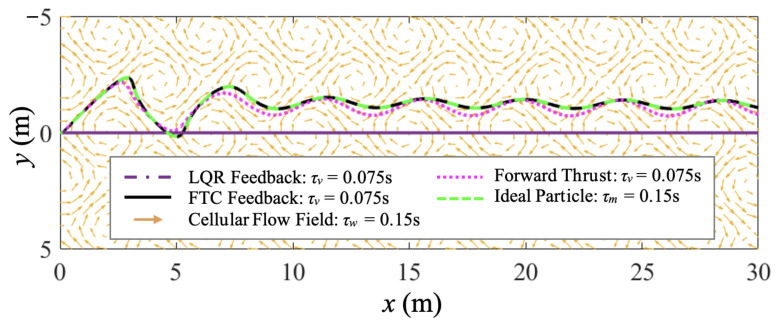
Case study 2: the trajectory comparison of the ideal fast-tracking particle with $\tau_{m}=0.15$ s, and purely thrust-driven, FTC-controlled, and LQR-controlled vehicles with $\tau_{v}=0.075$ s traversing a cellular flow with $\tau_{w}=0.15$ s.

**Figure 7 biomimetics-07-00192-f007:**
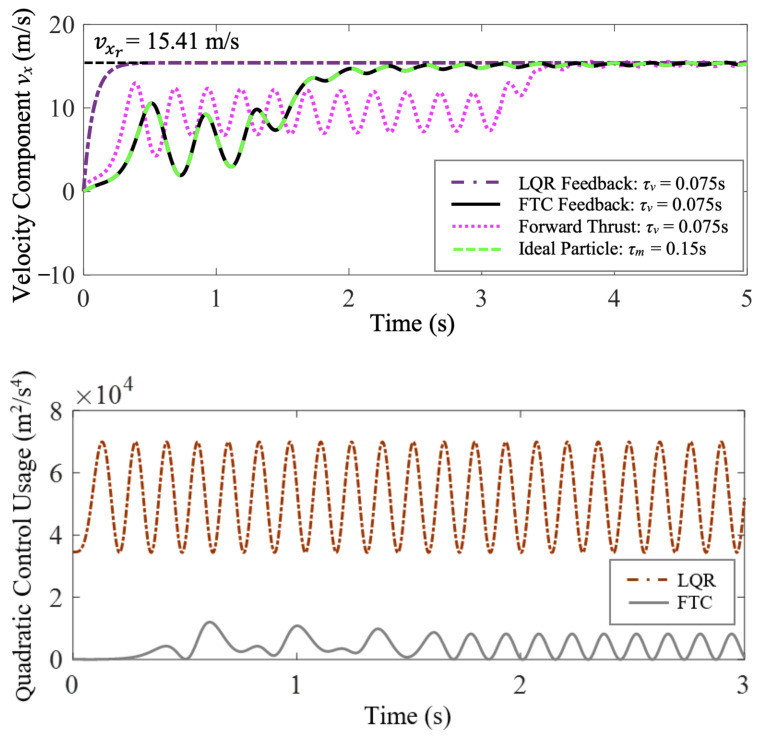
Case study 2: the comparison of the horizontal velocity of the ideal particle and purely thrust-driven, FTC-controlled, and LQR-controlled vehicles and the comparison of the FTC and LQR control cost show that the two controlled vehicles both achieve and maintain the desired horizontal steady-state velocity, but the FTC-controlled vehicle costs much less control effort.

**Figure 8 biomimetics-07-00192-f008:**
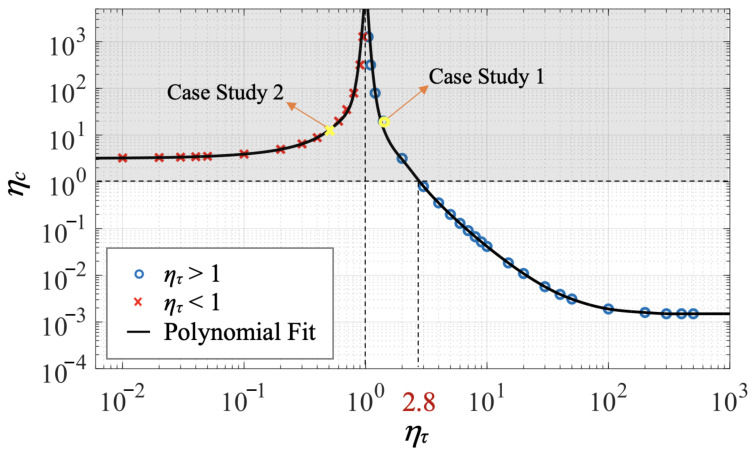
Log-log plot showing the relative total control cost ($\eta_{c}$) as a function of the normalized inertial response time of the vehicle ($\eta_{\tau }$) for $\tau_{m}=\tau_{w}$. Circular and cross markers represent simulations corresponding to different $\eta_{\tau }$, and the black lines are polynomial fits to these data points. Yellow markers represent the two simulations chosen as case studies in this section. Data points located in the grey shaded area correspond to simulations in which FTC costs less control effort than LQR.

**Figure 9 biomimetics-07-00192-f009:**
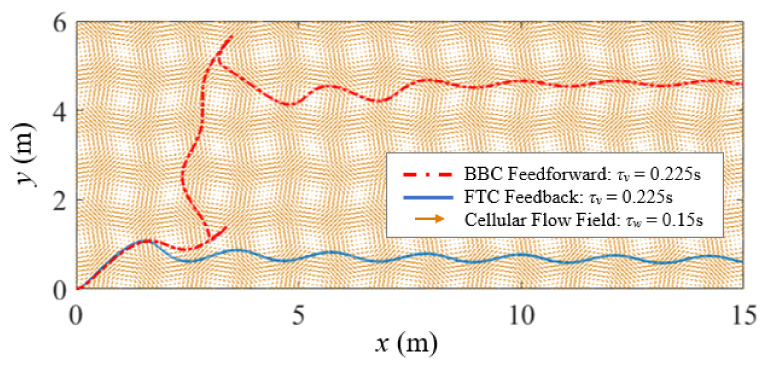
Case study 3: the trajectory comparison of the FTC-controlled and BBC-controlled vehicles with $\tau_{v}=0.225$ s traversing a cellular flow with $\tau_{w}=0.15$ s demonstrates that both vehicles achieve the control objective of reaching a desired horizontal position, $x_{d}=15$ m.

**Figure 10 biomimetics-07-00192-f010:**
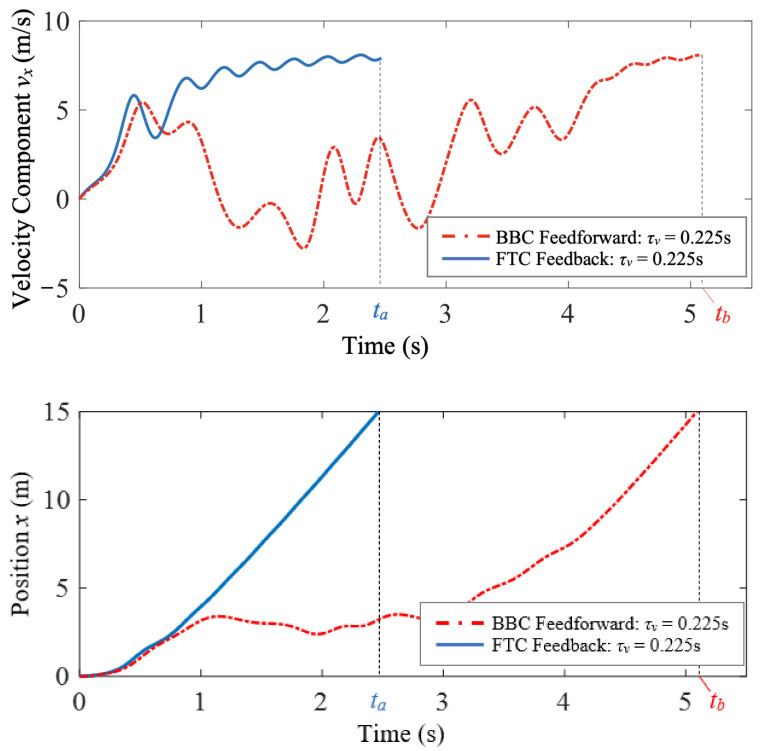
Case study 3: the comparison of horizontal velocity and position time histories of the FTC-controlled and BBC-controlled vehicles shows that the FTC-controlled vehicle reaches the desired horizontal position much faster than the BBC-controlled one.

**Figure 11 biomimetics-07-00192-f011:**
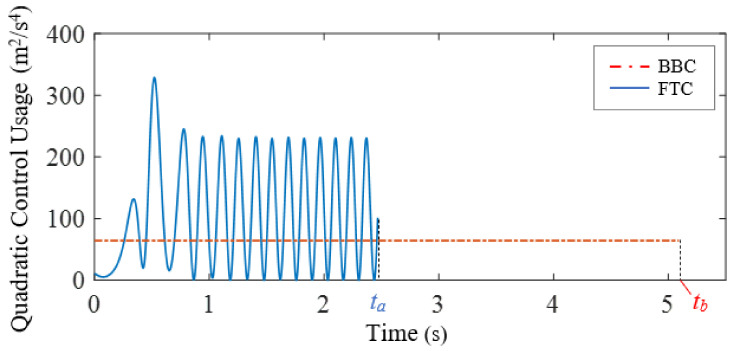
Case study 3: the comparison of the FTC and BBC quadratic control usage.

**Figure 12 biomimetics-07-00192-f012:**
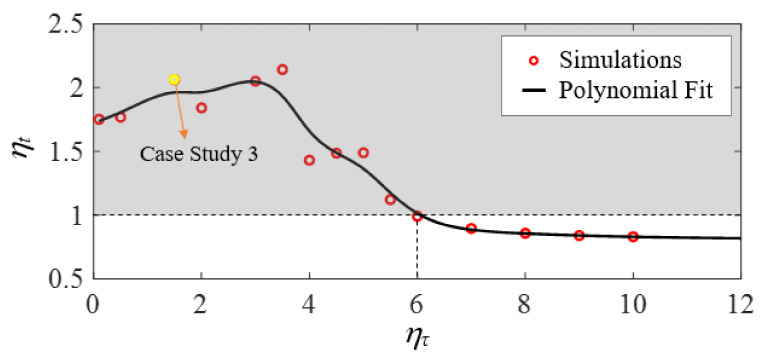
The ratio of total time ($\eta_{t}$) as a function of the normalized inertial response time of the vehicle ($\eta_{\tau }$) for $\tau_{m}=\tau_{w}$ is shown by performing many simulations (circular markers) corresponding to different values of $\eta_{\tau }$, and by performing a polynomial fit (black line) demarking case studies in which FTC uses less time than BBC (grey shaded area).

**Table 1 biomimetics-07-00192-t001:** Case study 3: the comparison of FTC and BBC performance.

Controller	$t_{f}$ (s)	$x(t_{f})$ (m)	*C* (m^2^/s^3^)
FTC	2.47	15.00	266.73
BBC	5.09	15.00	325.89

## Data Availability

Not applicable.

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
