# Peer review of "A Fast-Tracking-Particle-Inspired Flow-Aided Control Approach for Air Vehicles in Turbulent Flow"

_biomimetics, 2022, doi:10.3390/biomimetics7040192_

Round 1

Reviewer 1 Report

The authors propose a flow-aided control approach for air vehicles in turbulent flows inspired by the physics of fast-tracking particles in turbulent flows. The basic principle is straightforward: to move to the down-sweeping side of the vortices to take advantage of the tailwind and avoid the headwind. The control approach is implemented by minimizing the difference in acceleration between the vehicle and an ideal implicit particle model. The authors demonstrate, through a few case studies, that the proposed approach shows superior performance over previous approaches such as linear-quadratic regulator and the bang-bang control theory.

Congratulations to the authors on a well-written manuscript. I only have some minor comments to make.

Page 4, "Importantly, the normalized increase in settling velocity, ..., is approximately equal to one ..." I wonder if this is meant to be a rough estimate and if the precise value is not important. Because from Figure 1(b), the maximum is, apparently, at 0.3 rather than 1.

In equation (8). Is B a 2x2 identity matrix? How is it differentiated from B in equation (16) and later equations?

Page 8, "where the subscript m refers to "model" ..." I suggest that this be moved forward a bit where the subscript "m" first appears in xm.

Page 8, "because it can be assumed that the Reynolds number is smaller than one". How could the control approach be applicable to air vehicles if the ideal particle model operates in such a low Reynolds number regime?

Page 10, "both of which are purely driven by the same constant forward thrust" Does this exclude the control input?

Reviewer 2 Report

The paper presents a vehicle control strategy inspired by the particle fast tracking effect in homogeneous turbulence. The approach is demonstrated for cellular flow where it performs well compared to other control approaches. The paper covers an interesting topic and is well written and thorough. I have some questions/comments below:

1. Page 4 figure 1b and line 131. It is mentioned that the increased settling velocity is maximum when the settling parameter is approximately 1. Doesn’t fig 1b suggest it us closer to order 0.1? The peak appears to be at approximately 0.3.

2. Section 3 first paragraph. The analysis assumes that that the vehicle can be approximated by a point mass. I appreciate that this is common practice in the literature, but it would be useful to know how applicable this is for a real vehicle. For example, what approximate values for L_w/D_v are encountered by small UAVS.

3. How does the forward thrust approach perform for case study 2?

4. Some of the text is recycled between case study 1 and 2 which can be a bit repetitive. The text in case study 2 could be made more concise.
